# Coronary CTA Amidst the COVID-19 Pandemic: A Quicker Examination Protocol with Preserved Image Quality Using a Dedicated Cardiac Scanner

**DOI:** 10.3390/diagnostics13030406

**Published:** 2023-01-22

**Authors:** Alexisz Panajotu, Milán Vecsey-Nagy, Ádám Levente Jermendy, Melinda Boussoussou, Borbála Vattay, Márton Kolossváry, Örs Zs. Dombrády, Csaba Csobay-Novák, Béla Merkely, Bálint Szilveszter

**Affiliations:** 1Heart and Vascular Center, Semmelweis University, 1122 Budapest, Hungary; 2Gottsegen National Cardiovascular Center, 29. Haller Street, 1096 Budapest, Hungary; 3Physiological Controls Research Center, University Research and Innovation Center, Óbuda University, Bécsi út 96/b, 1034 Budapest, Hungary

**Keywords:** CT angiography, COVID-19, image quality, coronary artery disease

## Abstract

There has been an ongoing debate on the means to minimize the time patients spend at health care providers during the COVID-19 pandemic. We propose a strategy relying solely on intravenous (i.v.) beta-blocker administration for heart-rate (HR) control prior to coronary CT angiography (CCTA). We aimed to assess a potential difference in CCTA image quality (IQ) after implementation of a modified strategy compared to our standard protocol of oral premedication during the first wave of COVID-19. We analyzed CCTA examinations conducted one year before (*n* = 1511) and after (*n* = 1064) implementation of this new regime. Examinations were performed both on our 256-slice multidetector CT (MDCT) and dedicated cardiac CT (DCCT) scanners. We used a four-point Likert scale (excellent/good/moderate/non-diagnostic) for IQ assessment of the coronaries. We detected a significant increase in mean HR during examinations on both CT scanners (MDCT: 62.4 ± 10.0 vs. 65.3 ± 9.7, *p* < 0.001; DCCT: 61.7 ± 15.2 vs. 65.0 ± 10.7, *p* < 0.001). The rate of moderate/non-diagnostic IQ significantly increased on the MDCT (192/1005, 19.1% vs. 144/466, 30.9%, *p* < 0.001), while this ratio did not change significantly on the DCCT (62/506, 12.3% vs. 84/598, 14.0%, *p* = 0.38). The improved temporal resolution of DCCT allows the stand-alone use of i.v. premedication with preserved IQ; hence, the duration of visits can be shortened.

## 1. Introduction

During the first wave of the coronavirus disease 2019 (COVID-19) pandemic, a multitude of elective procedures and examinations were postponed or cancelled due to healthcare related restrictions and lockdowns [1]. There was an ongoing debate in almost every national and international society on how each provider can resume operation in the fundamentally changed situation [2]. Coronary CT angiography (CCTA) is a fast, non-invasive examination that plays a central role in the cardiac care of patients with chronic coronary syndrome [3,4]. According to the 2019 guideline of the European Society of Cardiology, CCTA was introduced as one of the first-line imaging modalities for patients with stable chest pain [5,6]. Following the outbreak of the pandemic, utilization of cardiac imaging shifted from procedures representing a higher risk of contamination to safer alternatives. The pandemic further increased the use of CCTA as an alternative to invasive coronary angiography (ICA) among patients presenting with chronic coronary syndrome, or to transesophageal echocardiography (TEE) before ablation/cardioversion procedures to evaluate the left atrial appendage [7].

Proper heart rate (HR) control is fundamental in order to achieve diagnostic image quality (IQ) for most of the patients by avoiding motion artifacts due to limited temporal resolution of the modality [8]. Recently, a dedicated cardiac scanner was introduced to overcome this limitation of single-source CT systems [9]. Appropriate HR control of patients before CCTA examination with oral beta-blockers was recommended as early as 2006, however these protocols suggested monitoring over 1 h after the administration of the medication [10]. It is well described, that the plasma peak concentration of the widely used metoprolol is reached at 90 min, whereas the peak effect of intravenous (i.v.) metoprolol occurs as early as 5–10 min after injection [11,12]. Our institutional protocol preceding the pandemic era recommended a 60 min wait after the initial dose of oral metoprolol. Such a regime of beta-blocker administration before CCTA examinations is considerably time-consuming, which in turn may increase the risk of COVID-19 dissemination. 

To reflect the safety measures demanded by the pandemic setting, beyond the standard precautions, our department also implemented a modified premedication strategy consisting of solely i.v. HR control prior to CCTA to shorten patient waiting time, thus potentially increasing patient and medical personnel safety. Although the safety of i.v. beta-blocker administration as premedication for CCTA has been documented before [13,14], there is scant evidence about its effect on IQ. This study aims to determine whether the implementation of this novel strategy influences CCTA IQ as compared to our previous standard premedication protocol.

## 2. Materials and Methods

### 2.1. Study Population

In our retrospective study, we analyzed CCTA examinations conducted one year before (standard protocol period—15 March 2019–14 March 2020) and after the implementation of the new premedication strategy (modified protocol period—15 March 2020–15 March 2021). The data of those consecutive patients who were examined in our institution were analyzed for the assessment of suspected coronary artery disease (CAD). Exclusion criteria from the analysis were cardiac arrhythmia or pacemaker dependency, and decreased IQ due to other reasons than motion artifacts. Due to the retrospective nature of the study, informed consent was waived by the institutional review board.

### 2.2. Heart Rate Control Using Beta-Blockade

The guidelines for the procedure of CCTA examinations recommend oral beta-blocker administration if the patient’s resting HR is above 60 beats per minute (bpm) in order to achieve optimal IQ [12]. Cardiac motion is still a limiting factor with most available CT scanners and reducing HR is one of the most important means of reducing motion artifacts in CCTA images. Our standard institutional protocol generally follows this recommendation by administering 25–100 mg of oral metoprolol 1 h prior to the scan if the patient has an HR above 65 bpm upon arrival. This is followed by the administration of i.v. metoprolol during the scan if the target HR is still not reached (5 mg stepwise doses, up to a maximum of 20 mg). Such a gradual administration of oral and i.v. beta-blockers before CCTA is considerably time-consuming which may increase the risk of further COVID-19 dissemination [10,11]. 

Shortly after the outbreak of the pandemic, we changed the premedication protocol in our institution (15 March 2020). Compared to the combination of oral and i.v. beta-blocker therapy used in the pre-pandemic protocol, our modified protocol utilizes solely i.v. premedication (Figure 1). In case of no contraindication, sublingual nitrates are routinely administered to every patient who undergoes CT imaging for the evaluation of CAD. 

### 2.3. CT Image Acquisition and Reconstruction

We performed CCTA image acquisition according to our institutional protocol. With both CT scanners, the first step is a surview of the chest, and afterwards native imaging comprising the heart is performed. HR was measured before and after the native images to evaluate the need for further i.v. beta-blocker administration. Contrast-enhanced CT angiography acquisition using step-and-shoot axial mode with ECG triggering was performed with our 256-slice multidetector CT (MDCT) (Brilliance iCT 256, Philips Healthcare, Best, the Netherlands) or dedicated cardiac CT (DCCT) (CardioGraphe, GE Healthcare, Chicago, IL, USA) scanners in both years. Tube voltage and tube current were adjusted according to patient body size. Post-threshold-scan delay was determined using a bolus tracking method with a threshold of 130 HU, and the region of interest was placed in the left atrium. Images were obtained after injecting 80–90 mL iomeprol contrast medium at a 4.5–5.5 mL/s flow rate (Iomeron 400; Bracco Imaging, Milan, Italy), preceded by 10 mL saline pacer and followed by 40 mL saline chaser bolus [15]. Reconstructions were performed using filtered back projection (FBP), hybrid iterative (iDose) and model based iterative (IMR, ASiR) algorithms, according to the vendors. All datasets were reconstructed with a standard kernel routinely applied in clinical practice (XCC for the MDCT scanner, CV standard for the DCCT). The effective radiation dose was calculated from the dose-length product of the scans with a k conversion factor of 0.014 (mSv per mGy cm) [16]. 

### 2.4. Image Quality Analysis

Subjective IQ was evaluated using the best reconstruction phases (predominantly end diastolic 78–81%) selected by the examiners. IQ was graded on a four-point Likert scale considering the degree of motion artifacts affecting the evaluation of the coronaries: (1) non-diagnostic, with severe motion artifacts seriously impairing evaluation; (2) moderate, with considerable motion artifacts, resulting in an IQ only sufficient to rule out significant luminal stenosis; (3) good, with preserved ability to assess the degree of lumen stenosis and (4) excellent, with no visible motion artifacts and sharp vessel contours (Figure 2). Interpretability was defined on a per-patient basis: if an evaluated coronary artery was rated as non-diagnostic, the corresponding patient was considered non-interpretable. The readers were instructed to ignore issues that could not be ascribed to the presence of motion artifacts (e.g., prominent image noise, poor contrast, extensive calcification or step artifacts). IQ assessment was performed as part of a clinical reading by four experienced cardiovascular radiologists (with >8 years of experience in reporting CCTAs). Assessment was carried out in each case with respect to our institutional standards. Reliability of the assessments is ensured by regular monthly consensus reads. Reproducibility between readers was assessed using a sample of 20 datasets to calculate Fleiss’ kappa value.

## 3. Results

In the first year of our study period, we analyzed 1511 adult patients (age: 58.1 ± 12.3, females: 45.0%), while 1064 patients were assessed in the second year (age: 57.7 ± 12.2, females: 41.4%). Examinations were performed on our two CT scanners (conventional 256-slice Philips Brilliance iCT (MDCT); new generation 560-slice GE CardioGraphe (DCCT)) and were analyzed separately. Basic demographic parameters and cardiovascular risk factors are summarized in Table 1. There was no significant difference in demographic parameters, however, we detected a higher rate of smokers and patients with diabetes on the MDCT arm and more patients with dyslipidemia on the DCCT arm before implementing the modified protocol. Other risk factors showed no significant difference. 

All readers re-evaluated 20 datasets in order to assess inter-observer variability. An observed kappa value of 0.81 was considered a measure of reliability, highlighting the excellent reproducibility of our evaluations.

Overall, 33.5% (406/1511) of the scans were performed on DCCT using the standard protocol, while this proportion significantly increased to 56.2% (466/1064) in the second year (*p* < 0.001). The average-per-patient dosage of i.v. metoprolol increased from 2.3 mg (SD: 3.96) to 4.2 mg (SD: 4.51) (*p* < 0.001). Premedication data is summarized in Table 2.

Comparing data we gathered from the standard and the modified protocol, we observed a significant difference in the mean HR during examinations using both devices (MDCT: from 62.4 ± 10.0 to 65.3 ± 9.7, *p* < 0.001; DCCT: from 61.7 ± 15.2 to 65.0 ± 10.7, *p* < 0.001). The target HR (<65 bpm) was achieved at a significantly lower rate in the second year using both devices (MDCT: 673/1005, 67.0% vs. 251/466, 53.9%, *p* < 0.001; DCCT: 363/506, 69.8% vs. 324/598, 54.2%, *p* < 0.001). The rate of moderate IQ and non-diagnostic scans significantly increased using MDCT (192/1005, 19.1% vs. 144/466, 30.9%, *p* < 0.001), while this ratio did not change substantially using DCCT (62/506, 12.3% vs. 84/598, 14.0%, *p* = 0.38). The rate of non-diagnostic scans alone did not change significantly with either scanner (MDCT: 29/1005 (2.9%) to 15/466 (3.2%), *p* = 0.73, DCCT: 11/506 (2.2%) vs. 9/598 (1.5%), *p* = 0.41). Image quality results are summarized Table 3.

## 4. Discussion

In our study we examined the effect of a premedication protocol solely using i.v. beta-blocker compared to the standard strategy using combined oral and i.v. beta-blocker administration on CCTA IQ. We observed that the IQ did not deteriorate significantly when using the newest generation device dedicated to CCTA, while using a 256-slice single source MDCT scanner, IQ was affected by motion artifacts, which resulted in a significant decrease in the ratio of excellent/good to moderate/poor quality scans.

The COVID-19 pandemic has markedly disrupted the delivery of healthcare services and therefore the life of patients worldwide. A global survey conducted by the International Atomic Energy Agency in 846 centers of 106 participating countries reported a reduction in testing as high as 70–80% in certain modalities. The impact of these changes is not yet fully understood and there are serious concerns about long-term adverse outcomes resulting from the decrease in diagnostics and treatment [17]. The same tendency was reported in case of elective CCTA examinations. Multiple North American cardiac centers deferred a significant proportion of elective imaging studies during the peak of the initial surge of COVID-19 infections and concentrated on the most urgent examinations and scans performed as preprocedural planning [4]. On the other hand, the role of cardiac CT examinations also evolved during the pandemic and CT was successfully used to solve diagnostic difficulties that normally would have required other modalities such as TEE, ICA or cardiac MRI. By using modified protocols for comprehensive cardiopulmonary assessment, CT can provide valuable information on cardiac and lung involvement, as well. In a recently published case report, CT was used to confirm the diagnosis of acute myocarditis-related myocardial injury as well as to rule out interstitial pneumonia, pulmonary embolism and coronary artery disease with a so-called “quadruple-rule-out” protocol [18]. A comprehensive collection of recommendations published by the European Society of Cardiology summarizes the potential of cardiac CT examinations in several bullet points: CCTA might be the preferable modality in patients with suspicion of symptomatic CAD as well as in certain acute chest pain cases, because it can be performed more rapidly with limited exposure of both medical personnel and other patients. Additionally, cardiac CT can be used instead of TEE for the rule-out of intracardiac thrombi. Furthermore, CCTA can be utilized in low-risk patients with non-ST segment elevation myocardial infarction and inconclusive troponin/ECG changes for quick risk stratification and an early discharge compared to a more time-consuming invasive approach [19,20]. In patients presenting with myocarditis, CCTA should be utilized to exclude CAD [4]. All these data underline that the indications and recommended protocols for examinations are rapidly evolving. The need for the development of pandemic-compatible clinical protocols and practices is obvious. 

It is important to emphasize the relevance of IQ in CCTA imaging as it is not enough to rule out or prove the presence of an obstructive atherosclerotic lesion, but the burden of non-obstructive CAD itself is also important. The results of both large observational studies and smaller clinical trials have pointed out that atherosclerotic plaque burden as well as certain plaque features represent risk factors and are predictors of later major cardiac events [21,22]. Certain adverse plaque characteristics, such as the napkin-ring sign and low-attenuation plaque were reported as the most important predictors of major adverse cardiac events [23]. For precise assessment of such vessel wall abnormalities, however, it is essential to have motion-artifact free, high-IQ datasets.

Beta-blocker administration is still necessary as a premedication for CCTA examinations. Even with the most advanced CT devices, HR control has a positive impact on IQ [24]. The coronary most frequently affected by motion artifacts is the right coronary artery [25], which correlates with our current results. Our data suggest though, that with a dedicated wide-detector array scanner the most commonly used relatively time-consuming oral metoprolol premedication can be substituted with solely i.v. administration of the substance. This new practice can greatly decrease the total time necessary for the examination, as the otherwise recommended 30–60 min wait for the effect of oral beta-blockers can be refrained from. A very thorough analysis of different premedication agents was published by Maffei et al. in 2009, in which the quickest and most effective HR reduction could be detected in patients treated with i.v. beta-blockers. From the overall study population, an HR of under 65 bpm and 60 bpm could be achieved in 81.8% and 56.7% of the cases, respectively. The average time required for premedication was reported as 44 ± 25 min, whereas the time necessary for solely i.v. premedication was significantly shorter, 8 ± 9 and 8 ± 8 min in two study groups [26]. The decreased time needed for preparation before the scan in the pandemic setting can decrease the risk of contamination in the imaging facility. 

The issue of beta-blocker non-responder patients should be addressed as well. One publication in the literature states that as much as 58% of patients who received oral metoprolol did not achieve the target HR prior to CCTA despite additional i.v. beta-blocker administration [27]. Maffei et al. reported an overall 18.3% non-responder rate in their investigation [26]. This phenomenon may affect our study population as well, but supposedly at equal rates in all study periods and on both scanners, thus it should not have an impact on our overall results. Nevertheless, further examination of metoprolol non-responders may warrant its own study in the future. 

The safety of beta-blocker premedication prior to CCTA examinations has been examined in several other studies. Maffei et al. also reported four adverse events in the examined five hundred and sixty patients, all of which were attributed to other factors than the premedication [26]. The safe applicability of esmolol, oral metoprolol and oral plus i.v. metoprolol was reported by several other authors [28,29,30]. During our own study period, there was no adverse event requiring hospital admission due to the administration of oral or i.v. metoprolol. 

With the advancement and evolution of CT technology, the first purpose-built cardiovascular CT scanner was introduced in 2017. It is a small footprint scanner, which provides coverage of the whole heart in one axial image acquisition step in a single heart beat using two X-ray sources with overlapping cone beams. This is complemented by a maximal gantry rotation speed of 0.24 s to achieve comparable IQ to other contemporary MDCT scanners [31]. The application of a novel model-based adaptive filter (MBAF2) has the potential to further increase IQ by reducing image noise and improving signal-to-noise and contrast-to-noise ratios [32]. These technological advancements have the potential to provide adequate coronary imaging to a wide array of challenging patients, as presented in the current study.

The current study has limitations that should be considered: First, the subjective nature of the analysis introduces the possibility of bias into the investigation. It should be emphasized that although readers were blinded to information pertaining to the CT acquisition, it is nevertheless impossible to exclude the possibility that the type of scanner could be deduced from the datasets. Second, diagnostic accuracy was not evaluated by correlating our results to the gold standard ICA. Furthermore, although it is safe to assume that time spent at our institution is substantially decreased by refraining from oral premedication, no data were collected with regards to exact individual times spent at the clinic. Finally, it is important to mention that i.v. metoprolol is more expensive than oral metoprolol; however, at this point we did not perform a cost–effectiveness analysis of the use of the i.v. agents. 

## 5. Conclusions

The results of our study suggest that when using the newest generation of dedicated cardiac scanners, i.v. premedication alone might be sufficient to obtain adequate CCTA IQ; therefore, the duration of the examination protocol can be shortened. Conversely, we also detected that conventional multidetector CT scanners may still require oral beta-blocker therapy before the examination in order to achieve optimal IQ. 

## Figures and Tables

**Figure 1 diagnostics-13-00406-f001:**
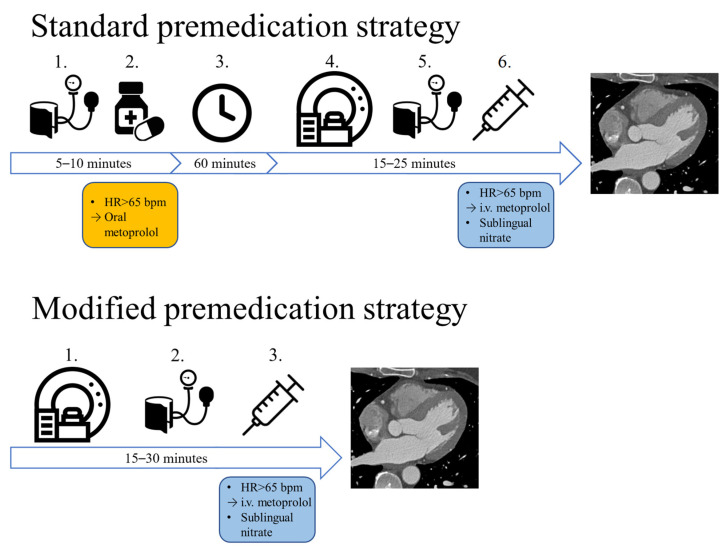
The two different premedication strategies: with the standard approach, if the patients’ heart rate is above 65 bpm, oral metoprolol is administered, which is followed by a 1 h wait. If afterwards the HR is still not optimal, i.v. metoprolol is administered right before image acquisition. With our modified premedication strategy, no oral medication is administered, only i.v. metoprolol is administered before the CTA scan. HR: heart rate, bpm: beats per minute, i.v.: intravenous.

**Figure 2 diagnostics-13-00406-f002:**
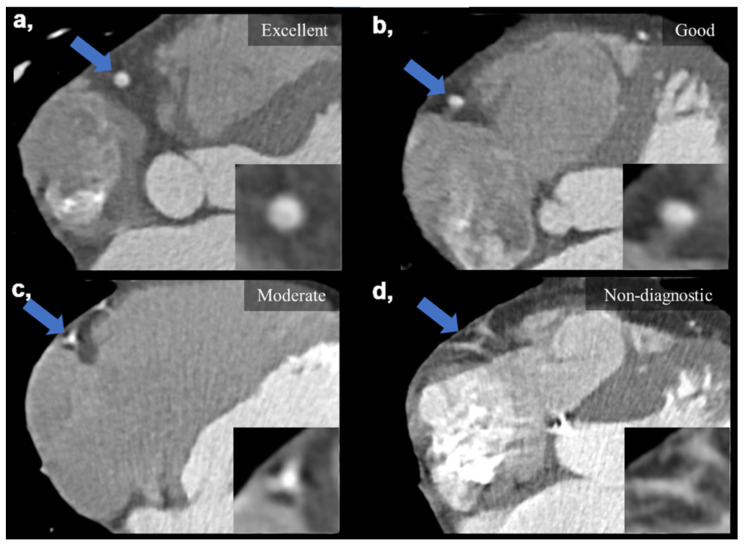
Visual representation of image quality categories on four example images taken on the DCCT. (**a**) Excellent IQ with sharp vessel contours and three grades of motion artifacts degrading the evaluation of the right coronary artery (**b**–**d**). (**b**) Good IQ—slight motion artifact not affecting luminal assessment. (**c**) Moderate IQ—only the presence of significant stenosis can be ruled out. (**d**) Non-diagnostic IQ—no luminal assessment is possible. Blue arrows mark the middle segment of the right coronary artery, which is also enlarged in the right lower corners of the images. DCCT: dedicated cardiac CT, IQ: image quality.

**Table 1 diagnostics-13-00406-t001:** Demographic parameters and cardiovascular risk profile of enrolled patients.

	Philips Brilliance iCT			GE CardioGraphe		
	Standard Protocol	Modified Protocol	*p*-Value	Standard Protocol	Modified Protocol	*p*-Value
Cases	1005	466		506	598	
Demographic data:						
Age (years)	58.0 ± 12.4	58.7 ± 12.0	0.32	58.3 ± 12.0	57.0 ± 12.3	0.07
Female, *n* (%)	443 (43.6)	187 (40.1)	0.15	237 (46.8)	254 (42.5)	0.15
BMI (kg/m^2^)	28.0 ± 4.8	28.1 ± 4.9	0.58	27.8 ± 4.7	27.8 ± 4.6	0.88
CV risk factors:						
Previous AMI, *n* (%)	72 (7.1)	27 (5.8)	0.33	22 (4.3)	17 (2.8)	0.18
Smoking, *n* (%)	365 (35.9)	132 (28.3)	0.003	183 (36.2)	223 (37.3)	0.70
Hypertension, *n* (%)	620 (61.0)	275 (59.0)	0.33	293 (57.9)	363 (60.7)	0.35
Diabetes, *n* (%)	141 (13.9)	48 (10.3)	0.047	68 (13.4)	70 (11.7)	0.39
Dyslipidemia, *n* (%)	396 (38.9)	186 (39.9)	0.85	222 (43.9)	226 (37.8)	0.04

Continuous variables are expressed as mean ± standard deviation (SD), while categorical variables are expressed as numbers and percentages. BMI: body mass index, CV: cardiovascular, AMI: acute myocardial infarction.

**Table 2 diagnostics-13-00406-t002:** Premedication and imaging parameters.

	Philips Brilliance iCT			GE CardioGraphe		
	Standard Protocol	Modified Protocol	*p*-Value	Standard Protocol	Modified Protocol	*p*-Value
Cases	1005	466		506	598	
Oral metoprolol (mg):						
0, *n* (%)	308 (30.6)	-		127 (25.1)	-	
25, *n* (%)	89 (8.9)	-		49 (9.7)	-	
50, *n* (%)	291 (29.0)	-		133 (26.3)	-	
75, *n* (%)	175 (17.4)	-		91 (18.0)	-	
100, *n* (%)	142 (14.1)	-		104 (20.6)	-	
Intravenous metoprolol (mg):			<0.001			<0.001
0, *n* (%)	635 (63.2)	182 (39.1)		396 (78.2)	274 (45.8)	
5, *n* (%)	229 (22.8)	159 (34.1)		76 (15.0)	224 (37.4)	
10, *n* (%)	117 (11.6)	91 (19.5)		28 (5.5)	88 (14.7)	
15, *n* (%)	20 (2.0)	25 (5.4)		6 (1.2)	12 (2.4)	
20, *n* (%)	4 (0.4)	9 (1.9)		0 (0.0)	3 (0.5)	
Effective radiation dose (mSv):	5.3 ± 2.6	5.4 ± 2.0	0.33	3.6 ± 1.2	4.0 ± 1.8	<0.001

Continuous variables are expressed as mean ± standard deviation and categorical variables are expressed as numbers and percentages. mSv: millisievert.

**Table 3 diagnostics-13-00406-t003:** Comparison of image quality using the different heart rate protocols.

	Philips Brilliance iCT			GE CardioGraphe		
	Standard Protocol	Modified Protocol	*p*-Value	Standard Protocol	Modified Protocol	*p*-Value
Cases	1005	466	-	506	598	-
Heart rate (bpm)	62.4 ± 10.0	65.3 ± 9.7	<0.001	61.7 ± 15.2	65.0 ± 10.7	<0.001
Target heart rate < 65 bpm, *n* (%)	673 (67.0)	251 (53.9)	<0.001	363 (69.8)	324 (54.2)	<0.001
Overall image quality:			<0.001			0.38
Excellent/good	813 (80.9)	322 (69.1)		444 (87.7)	514 (86.0)	
Moderate/non-diagnostic	192 (19.1)	144 (30.9)		62 (12.3)	84 (14.0)	
Non-diagnostic scan, *n* (%)	29 (2.9)	15 (3.2)	0.73	11 (2.2)	9 (1.5)	0.41

Continuous variables are expressed as mean ± standard deviation and categorical variables are expressed as numbers and percentages. bpm: beats per minute.

## Data Availability

The data presented in this study are available on request from the corresponding author. The data are not publicly available due to reasons pertaining to patient privacy.

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
