# Peer review of "Coronary CTA Amidst the COVID-19 Pandemic: A Quicker Examination Protocol with Preserved Image Quality Using a Dedicated Cardiac Scanner"

_diagnostics, 2023, doi:10.3390/diagnostics13030406_

Round 1
Reviewer 1 Report
Synopsis
The authors aimed to evaluate two premedication protocols (oral + i.v. metoprolol vs i.v. metoprolol alone) for CCTA. They analyzed CCTA examinations conducted using the combined protocol (n=1511) vs. the i.v. protocol (n=1064), acquired either on a 256-slice MDCT or a dedicated cardiac CT (DCCT) system. A four-point Likert scale (excellent/good/moderate/non-diagnostic) was used for image quality assessment. The rate of moderate/non-diagnostic IQ significantly increased on MDCT using the i.v. protocol (192/1005, 19.1% vs. 144/466, 30.9%, p<0.001), while no such change was observed on DCCT (62/506, 12.3% vs. 84/598, 14.0%, p=0.38). They concluded that the improved temporal resolution of DCCT allows the stand-alone use of i.v. premedication with preserved image quality.
Comments:
-Overall well written paper with potential impact on patient management
-Was there a 33% drop in CCTA exams pre vs post pandemic?
-There is also an interesting drop in scanner utilization. While the exam rate on the Philips scanner dropped to half pre vs post pandemic, the rate increased on the CardioGraphe. Please explain
-Please indicate who evaluated image quality with what level of experience. How many readers assessed each dataset? Subjective evaluation is typically expected to be performed by multiple readers to provide inter-observer agreement. Please elaborate if this was done and provide details in the methods and results.
-Why was extensive calcification considered an exclusion criterion?
-I would suggest to merge Figs 1 and 2, maybe create a timescale (maybe 90 minutes) on the arrow, and display the two protocols above and below the arrow. Also, the second step in Fig 2 (BP/HR check) would also be repeated before i.v. administration in Fig 1
-Fig 3 - were these all done on one of the scanners?
-Not exactly sure how much this difference affects the results, but the size of the metoprolol dose groups are quite different; e.g. standard protocol i.v. 5mg 22.8% vs 15%; 10mg 11.6% vs 5.5% etc. Can you please comment on that?
-Were these mostly in or out patients? Because oral premedication in an inpatient setting would not count towards the "examination time"
-What's the post-administration surveillance requirement for patients with oral vs i.v. metoprolol? Does an outpatient require additional time to be monitored if received i.v. metoprolol?
Reviewer 2 Report
Panajotu et al. present a manuscript entitled „Coronary CTA amidst the COVID-19 Pandemic: a Quicker Examination Protocol with Preserved Image Quality Using a Dedicated Cardiac Scanner”. In this single-center study, the author compare patient preparation protocols for cardiac CTA and their influence on image quality in two different scanners. Rationale of the article is to reduce patient contact time in the COVID 19 pandemic by using a beta-blocker only strategy compared to oral medication. They found an increase in moderate or non-diagnostic image quality scans for a conventional CT scanner, but not for a dedicated cardiac CT scanner. They concluded that the advantages of the dedicated cardiac CT scanner concerning temporal resolution allow using stand-alone i.v. beta-blocker.
Overall, this is a well-written article with a clear message. I have only some minor comments to address by the authors:
- Title: ok
- Abstract: ok
- Keywords: please check with MESH database
- Introduction: well-written, comprehensive. One minor comment – reduced waiting time would also increase the safety of the medical staff, not only other patients.
- Methods: Please describe or present flow chart how many patients were excluded based on the exclusion criteria. Please describe who read the CT scans, how many readers were involved. Is there data about reproducibility?
- Otherwise: methods well explained
- Results: ok
- Discussion: I would avoid discussing vendor names specifically in the discussion, the vendor is already mentioned in the methods, otherwise well written
- Literature: ok
- Figures and Tables: ok
Reviewer 3 Report
The article is clearly structured and well written. However overall novelty is a bit questionable. There are two main messeages 1) using IV betablockers is faster then oral, 2) newer CT is better than older type.
1) Using oral and IV betablockers and other heart rate lowering medications is well know and works in rutine practice. Authors used max 20 mg IV metoprolol. Usually is recommended up to 15 mg but in some reports is used safely up to 30 mg. [1, 2] Based on experience from my hospital in chronic CAD could be used protocol when patient took 50-100 mg per os metoprolol at home in the morning and in most of these cases no further mediacation is needed at CT department.
2) Newer scanners bring usually faster scanning and with higher temporal resolution is possible to scan during higher heart rate. In the higher heart rate systole could be find as the best phase for coronary arteries. [3, 4]
Therefore my recommendation is: Minor revision
I think in the article should be mentioned also range of RR interval during aquisition. There is only mentioned that the best reconstruction phases were predominantly diastolic 78-81%. However there is no information whether systole was also available.
References:
[1] BFCR(14)16. Royal College of Physicians (RCP) and the British Society of Cardiovascular Imaging (BSCI), The Royal College of Radiologists (RCR). Standards of practice of computed tomography coronary angiography (CTCA) in adult patients. BFCR(14)16. 2014
[2] Pannu, H. K., Alvarez, W., & Fishman, E. K. (2006). β-blockers for cardiac CT: A primer for the radiologist. In American Journal 352 of Roentgenology (Vol. 186). doi: 10.2214/AJR.04.1944 - alraedy cited in the article
[3] Seifarth H, Wienbeck S, Püsken M, Juergens KU, Maintz D, Vahlhaus C, Heindel W, Fischbach R. Optimal systolic and diastolic reconstruction windows for coronary CT angiography using dual-source CT. AJR Am J Roentgenol. 2007 Dec;189(6):1317-23. doi: 10.2214/AJR.07.2711. PMID: 18029865.
[4] Le Roy J, Azais B, Zarqane H, Vernhet Kovacsik H, Mura T, Lacampagne A, Amedro P. Selection of optimal cardiac phases for ECG-triggered coronary CT angiography in pediatrics. Phys Med. 2021 Jan;81:155-161. doi: 10.1016/j.ejmp.2020.12.002. Epub 2021 Jan 15. PMID: 33461028.
